# Discovery and Extraction of Cultural Traits in Intangible Cultural Heritages Based on Kansei Engineering: Taking Zhuang Brocade Weaving Techniques as an Example

**Yuedi Huang [1,2] and Younghwan Pan [2,*]**

1   Department of Smart Experience Design, Guilin University of Technology, Guilin 541004, China;
    huangyuedi123@kookmin.ac.kr
2   Department of Smart Experience Design, Kookmin University, Seoul 02707, Korea
*   Correspondence: peterpan@kookmin.ac.kr; Tel.: +82-10-3305-1011

**Abstract:** Taking China's national intangible cultural heritage (ICH) Zhuang brocade as the research object, its cultural traits were extracted through scientific methods from the perspective of Kansei Engineering. Samples were collected through desk research, expert interviews, and questionnaires for qualitative and quantitative research. The semantic differential method was adopted to analyze the vocabulary descriptions of different types of Zhuang brocade samples, and the Likert scale was used to measure the viewer's visual perception. Eye tracker experiments were conducted to verify and further explore the cultural traits of Zhuang brocade so that the emotions in this ICH can be quantified more scientifically. Based on the heat map and data, scientific and reasonable descriptions and typical shapes best matching Zhuang brocade cultural traits were acquired. By using new technologies to interpret ICHs, this study proposed another way to extract cultural traits from ICH. The extracted Zhuang brocade cultural traits in this study could help improve the understanding of Zhuang brocade. This study could also provide certain references for the modern application and design of Zhuang brocade.

**Keywords:** intangible cultural heritage; Zhuang brocade; cultural trait; kansei engineering; eye tracker experiment

## 1. Introduction

Intangible cultural heritages (ICHs) are the shared wealth of all mankind. According to the UNESCO definition in the Convention for the Safeguarding of the Intangible Cultural Heritage, ICH means the practices, representations, expressions, knowledge, and skills—as well as the instruments, objects, artifacts, and cultural spaces associated therewith—that communities, groups and, in some cases, individuals recognize as part of their cultural heritage [1]. This ICH, transmitted from generation to generation, is constantly recreated by communities and groups in response to their environment, their interaction with nature and their history, and provides them with a sense of identity and continuity, thus promoting respect for cultural diversity and human creativity [2]. Benignly combining the live transmission of ICH with modern life has always been the focus of the academic circle and the desired outcome this study intended to reach. Weaving and embroidery fall into the ICH category of traditional crafts and are rare treasures in Chinese history. To a certain extent, weaving and embroidery reflect the diligence and wisdom of the working people, historical changes, and the evolution and development of national cultural customs, which endow them with great research value. By the time of writing, China's ICH protection efforts have entered the stage of consolidating the rescue and protection achievements and improving the inheritance practice. It is necessary to uphold the concept of integrating ICHs with the craftsmen, the objects, and daily life and continuously promoting the continuation, development, and revitalization of ICH in contemporary life [3].

The ICH of Zhuang brocade is the Zhuang brocade weaving techniques or Zhuang brocade in short. In 2006, the Zhuang brocade weaving techniques were included in the representative items of China's national ICHs [4]. Zhuang brocade, together with Nanjing brocade, Sichuan brocade, and Suzhou Song brocade, is famed as the four famous brocade in China and is the only minority brocade among them. Zhuang brocades are traditional silk fabrics of the Zhuang nationality in China and exquisite articles or handicrafts of everyday use woven with cotton or silk threads. Zhuang brocade has a long history and could be dated back to the early hemp weaving techniques in Guangxi, China, which served as the basis for its emergence [5]. Brocades emerged in Guangxi as early as in the Han Dynasty (202 BCE to 220 CE). The Zhuang brocade techniques took shape in the Tang and Song Dynasties (early 7th century to late 13th century) and further developed during the Ming and Qing Dynasties (1368 to 1912). In the Ming Dynasty (1368 to 1644), Zhuang brocade was listed as one of the articles of tribute. Zhuang brocade began to decline from the end of the Qing Dynasty to the beginning of the Republic of China (1840 to 1911) [6]. Over its history of more than a thousand years, Zhuang brocade has developed into a system of three types, more than 20 varieties, and more than 50 patterns. Zhuang brocade is famous for its durability, exquisite craftsmanship, unique designs, and beautiful patterns.

Every ICH item has its unique cultural traits. With relatively abundant materials today, consumers shifted from simple, functional consumption to emotional consumption [7]. The market demands more products with cultural heritage. If the unique cultural traits of ICH can be extracted and used in ICH-related design projects, the cultural heritage of the product can be enhanced to meet the emotional needs of the consumers. Therefore, this study attempts to introduce the cultural traits of ICH to the derivative products through kansei engineering. Taking advantage of the emotional quantification capacity of kansei engineering, the cultural traits of ICH were extracted and explored, which could inspire the design of ICH derivative products. The extracted cultural traits of Zhuang brocade could provide a theoretical basis for the development of relevant ICH derivative products, which is not only beneficial to the development of ICH derivative products but also the marketization of ICH.

## 2. Research Status, Background, and Purpose

### 2.1. Analysis of Research Status

The existing research in the field of ICH mainly focuses on the method of protection and the content of inheritance. In terms of protection, digital methods are the main focus of research; in terms of inheritance, the live inheritance and modern application of ICH are the focus of research [8]. Studies related to Zhuang brocade are mainly in the fields of folk culture, ethnology, historical development, technical procedures, and visual pattern modeling. Studies on the inheritance and application of Zhuang brocade culture mainly focus on cultural and creative product design, clothing design, and visual communication design. The study of cultural traits is rarely involved [9]. On the other hand, the research concerning the cultural traits in the field of product design focused on the design alone, and few covered the process from the cultural traits of ICH to their applications in cultural and creative products. In terms of research methodology, many studies focused on finding the point where the product meets the emotional needs of users from the perspective of user experience.

Kansei engineering was established in response to social development and market demand. In 2004, Donald Norman mentioned the relationship between emotion and design in Emotional Design [10]. Since 1970, kansei engineering has taken shape by considering the emotions and needs of the occupants in house design and has received more and more attention. In the 1990s, kansei engineering was widely implemented in the automobile industry [11]. Kansei engineering is a discipline that combines the engineering part of emotional quantitative measurement with the perception expression part on the basis of measurement, creation, and sensing. The core of kansei engineering is the measurement of human physiology and psychology. Technology is used in experiments

to more accurately quantify consumers' psychological indicators [12]. Kansei engineering aims to reduce the statistical errors caused by perceptual factors and bring the experimental results closer to consumers' psychological expectations. In 1989, Japanese scholar Mitsuo Nagamachi published a monograph entitled Kansei Engineering and constructed the theoretical system of kansei engineering in more detail [13]. The Ergonomics Laboratory of the University of Nottingham is a well-known research institution for Kansei Engineering. Research in this field has also been carried out in universities such as Hunan University and Zhejiang University in China. The basic research methods of kansei engineering in many studies include: combining psychology with sociology to study brain structure and neural transmission of perception, using modern computing technology to capture and study data of the users, and researching the design and manufacturing of products that meet the needs of users, i.e., kansei creative engineering [14]. By using kansei engineering methods to study the cultural traits of ICH, this study falls into the third category.

*2.2. Research Background*

With the development and evolution of society, the lifestyles and aesthetic tastes of the people have undergone great changes, which raises questions for the live inheritance of ICHs such as Zhuang brocade. Thus, integrating ICHs into modern life has become a problem that cannot be ignored. On the one hand, the current status of traditional Zhuang brocade is that it is no longer adapted to the development of society, gradually fading out of people's daily life, and drifting farther and farther away from modern aesthetic trends. On the other hand, the Chinese government has granted a lot of policy guidance and financial support to the inheritance and protection of ICH. In March 2017, the State Council of China passed the Plan on Revitalizing China's Traditional Crafts to promote the inheritance and development of traditional Chinese crafts, which provided a guarantee for the protection and development of ICH [15]. The inheritance of ICH has gradually entered the public's view and attracted widespread attention. With the continuous issuance of policies combining ICH with cultural and creative products, tourism, and poverty alleviation, ICHs have received more and more important cultural missions [16]. In recent years, Zhuang brocade production and protection demonstration bases have been established one after another in Guangxi Zhuang Autonomous Region, China. The appeal and guidance of the government have brought more and more attention to ICHs. Therefore, more and more companies and merchants are beginning to realize the increasing importance of consumers' emotional and psychological needs in product selection.

Holbrook and Hirschman (1982) pointed out in their classic work, the Experiential Aspects Of Consumption: Consumer Fantasies, Feelings, And Fun, that emotional and experiential consumption will come to the stage of history in place of traditional rational functional consumption. Holbrook put forward the 4Es consumer experience dimension and pointed out that in the era of increasingly abundant materials, consumers' choices are beginning to shift to consumption experience of the emotional level [16]. In Hauffe's experience design theory, consumers at different levels have different requirements for the products. Consumers at the viewer level require consumption at the aesthetic level (aesthetic experience) to obtain visual pleasure [17]. Therefore, the introduction of the design theory is especially important and urgent for the inheritance and modern application of ICH. From the perspective of the design theory, the development of cultural products and articles of daily life related to ICH has a positive significance for the live inheritance and promotion of ICH.

*2.3. Research Purpose*

In the market, more and more ICH-related products are put into R&D and mass production. From 2019 to 2021, more and more merchants and online platforms began to sell ICH-related products after taobao.com took the lead in launching the ICH products shopping festival. The ICH fever indicated the strong cultural demand in the market. In such a market environment, elements of Zhuang brocade are applied to many commercial

products. Cultural and creative products, articles of daily life, clothing, and decorations related to Zhuang brocade are presented to the consumers in a modern way. For example, Zhuang brocade handbags, Zhuang brocade coasters, Zhuang brocade scarves, Zhuang brocade cultural shirts, and other Zhuang brocade fashion products that are closer to modern life have become the favorites of the consumers. Therefore, consumers of cultural and creative products related to ICH are driven by their emotional needs, which are reflected in the cultural traits. Since Zhuang brocades are patterned objects, a lot of after-view emotions are included in the cultural traits. Visual perception and emotional quantification are also manifested in cultural traits. Most of the cultural traits of Zhuang brocade are derived from visual cognition, which makes them subjective emotions. Since emotions are usually invisible or difficult to express, their measurement and description are challenging. Therefore, group discussions and questionnaire surveys were adopted for the qualitative analysis and quantitative research on the words describing the collected cultural traits. Based on eye movement experiments, the cultural traits of Zhuang brocade were extracted, and the typical Zhuang brocade shapes best matching those cultural traits were identified, which could pave the way for the development and application of ICH products. This study is preliminary research on the design and development of ICH derivative products using kansei engineering. The main research questions of this study are as follows: (1) What are the cultural traits of the Zhuang brocade? (2) How to extract the cultural traits of ICH scientifically? (3) What are the positive effects of the extracted cultural traits on the inheritance and protection of ICH?

## 3. Design of the Research Process

Large number of Zhuang brocade pictures and documents were collected through on-site investigations, literature surveys, and desktop research. After removing materials of high similarity, the Zhuang brocade sample picture library was established.

The Zhuang brocades were visually categorized based on their appearance parameters, and the E-prime software was used to analyze and screen out the representative core samples.

Using expert interviews and the data from the first questionnaire survey, the vocabulary describing Zhuang brocade was extracted from the sample through statistical induction. The semantic differential method was adopted to analyze the vocabulary description of different Zhuang brocade patterns.

Based on user satisfaction and the demand transformation theory in interaction design, the core description vocabulary and adjective pairs were obtained through the second questionnaire, expert interviews, and desk research. Thus, the specific data of Zhuang brocade cultural trait description vocabulary were obtained from the viewers.

The relationship between Zhuang brocade description vocabulary and the samples was analyzed based on the third questionnaire survey data. Specifically, factor analysis and quadrant analysis in SPSS were adopted, of which factor analysis was adopted to extract the cultural traits, and quadrant analysis was adopted to classify the samples and the cultural traits.

Eye movement experiments were conducted to verify the cultural traits of Zhuang brocade obtained through questionnaire surveys and desk research. The colors and patterns of the Zhuang brocade samples were extracted based on heat maps to obtain the typical Zhuang brocade graphics most in line with their cultural traits. The specific research process is shown in Figure 1.

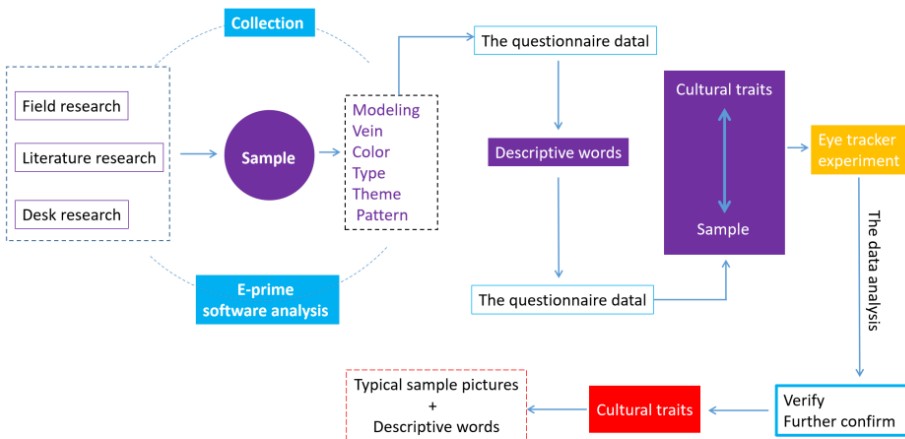

**Figure 1.** This is the research circuit.

## 4. The Extraction Process of Zhuang Brocade Cultural Traits

### 4.1. Collection of Research Samples

Through field inspections, image website browsing, museum collection browsing, and searching the photo library of China's ICH digital museum, a large number of valuable texts and pictures were obtained. The collection of sample pictures can be divided into three parts: on-site photography, museum photography, and Internet download. First of all, a large number of Zhuang brocade pictures were collected from Zhuang brocade-producing areas, including Jingxi County, Xincheng County, Binyang County, Longzhou County, and Daxin County using cameras. Thus, the Zhuang brocades currently sold and used in Guangxi were sampled through field research. Then, discussions on the digital measures for ICH protection were held with Ms. Tan Xiangguang, the ICH inheritor of Zhuang brocade weaving Techniques, in the Xiangguang Brocade Workshop in Binyang County, Guangxi. After that, the Anthropology Museum of Guangxi, the Museum of Guangxi Zhuang Autonomous Region, the National Museum of China, and the Palace Museum were visited to take pictures of the more exquisite collections of Zhuang brocades, which were used as important picture samples. Finally, more sample pictures of Zhuang brocade were collected from online platforms such as China's ICH digital museum [18], Guangxi intangible heritage protection website [19], and Baidu Pictures. The standard for picture collection is high definition, with a resolution of 300 or above. The main subjects of the photographs were the Zhuang brocades in sold, in use, and among museum collections. The collection was conducted as comprehensively and completely as possible. Thousands of Zhuang brocade pictures were collected. After excluding similar pictures, 400 samples of Zhuang brocade pictures were selected as the research objects, as shown in Figure 2.

Among the many pieces of literature about Zhuang brocades, different authors classified Zhuang brocades from different angles. For example, color, origin, function, theme, pattern, graphic style, and expression. Therefore, on the basis of different pieces of literature, the visual appearance variables of Zhuang brocade were classified as shown in Table 1. In Table 1, the variables were classified according to color, place of origin, theme, and pattern. The classification was based on the representative literature related to Zhuang brocade. The main reference for Zhuang brocade classification was the Study on Emblazonry of Zhuang Brocade by Dr. Lu of Shandong University [20]. The different classifications of Zhuang brocade in Zhuang Brocade of the Our Guangxi series by Wu Weifeng and Cai Hong were also used for reference [21]. Functions and specific pattern types were not used as variables because they were relatively complex and varied.

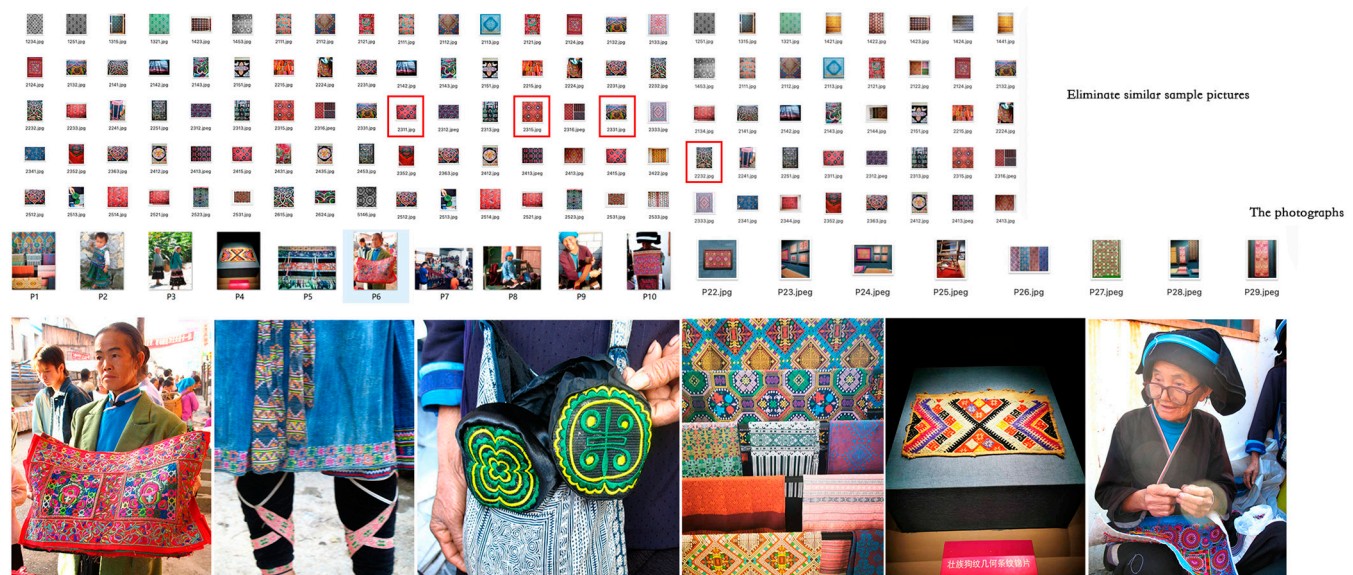

**Figure 2.** Part of the sample pictures and field trip pictures from the sample library.

**Table 1.** The division of Zhuang brocade visual appearance variables.

| No. | Color V1 | Origin V2 | Theme V3 | Graphic Style V4 |
|---|---|---|---|---|
| 1 | Plain brocade | Jingxi County, Guangxi | Animal themes | Four-direction continuous pattern |
| 2 | Colorful brocade | Xincheng County, Guangxi | Character theme | Two-direction continuous pattern |
| 3 | No | Binyang County, Guangxi | Text theme | Round fitted pattern |
| 4 | No | Longzhou County | Plant theme | Square fitted pattern |
| 5 | No | Daxin County | Myths and allusions | Separate pattern |

According to the classification in the above table, the 400 Zhuang brocade samples were coded to build the Zhuang brocade picture library. The first to the fourth digit of the code corresponds to V1, V2, V3, and V4, respectively. For example, the code of sample Z1 in the image library corresponds to colored brocade (2) under the color column (V1), Binyang County (3) under the origin column (V2), text theme (3) under the theme column (V3), and the four-direction continuous pattern (1) under the pattern column (V4). Therefore, the shape variable code of sample Z1 is 2331. In this way, the Zhuang brocades were first classified according to color, plain brocades were coded 1XXX, and colored brocades were coded 2XXX. Then, the brocades were classified according to the origin, the theme, and finally, the graphic style. Specifically, plain brocades are generally made by spinning a single-color yarn on a single-color cloth, and the background color is usually black or white. The two-direction continuous pattern means the shapes can continue in both left and right directions, which is usually found in lace. The four-direction continuous pattern means the shapes can continue in the up, down, left, and right directions, which is usually found in Zhuang brocades on clothes.

### 4.2. Selection of Research Samples

The 400 samples were coded according to the visual variables to establish a classified picture library, and the sample pictures were analyzed using the E-prime software. E-Prime is short for Experimenter's Prime (best). It is an experiment generation system for computerized behavior research jointly developed by Carnegie Mellon University, the Learning Research and Development Center at the University of Pittsburgh, and the Psychology Software Tools, Inc. in the United States [22]. In the analysis using the E-prime

software, 20 experts and 150 ordinary adults (male to female ratio of 1:1) were selected for the experiment. The experimenter selected the sample pictures that best match the cultural traits of Zhuang brocade according to their first visual experience. The 10 samples were selected based on the criteria of comprehensive subject matter, representative patterns, and common applications. Considering the analysis in E-prime, the sample pictures with cumulative frequency over 20 times and response time below 0.700 s were selected. Finally, 10 pictures of the core representative samples were determined, as shown in Table 2.

**Table 2.** Core representative sample pictures and their codes.

| Core sample | Sample Z1 | Sample Z2 | Sample Z3 | Sample Z4 | Sample Z5 |
|---|---|---|---|---|---|
| Pictures | | | | | |
| Code | 2331 | 2544 | 1511 | 2132 | 2441 |
| Core sample | Sample Z6 | Sample Z7 | Sample Z8 | Sample Z9 | Sample Z10 |
| Pictures | | | | | |
| Code | 2543 | 2152 | 2311 | 2412 | 1141 |

*4.3. The Collection and Selection of Descriptive Vocabulary of Zhuang Brocade Cultural Traits*

The extraction of Zhuang brocade cultural traits is indispensable to the scientific expression of its descriptive vocabulary. Scientifically describing the Zhuang brocade cultural traits is the prerequisite for their extraction. First, about 300 descriptive vocabularies of the Zhuang brocade cultural traits were obtained through desk research. Then, 50 more accurate words were obtained after classification and exclusion during group discussion. After that, 10 core vocabulary corresponding to the 10 core samples were obtained through a questionnaire survey. Finally, the semantic differential method and the Likert scale were adopted in the questionnaire. The sentiment quantification of the Zhuang brocade was measured through factor analysis of the test data.

### 4.3.1. Collection of Zhuang Brocade Descriptive Vocabulary

Using the desk research method, 20 craft artists, designers, and scholars of Zhuang brocade were invited to form the expert group. The expert group described the cultural traits of the Zhuang brocade from the perspectives of overall style, themes, patterns, and color characteristics of the 400 samples in the library and the 10 core samples. Finally, based on the literature of design and linguistics, the word meaning and vocabulary description was analyzed through the semantic material collection, semantic vocabulary synonym induction, and semantic vocabulary refining. The purpose was to qualitatively classify the collected descriptive vocabulary of Zhuang brocade cultural traits.

Through group discussion, 50 vocabularies were selected and assembled into a vocabulary list, as shown in Table 3.

**Table 3.** Vocabulary describing the Zhuang brocade cultural traits.

| Traits | Descriptive Vocabulary (Adjectives) of Zhuang Brocade Cultural Traits |
|---|---|
| Overall style | Fresh, plain, beautiful, bright, concise, ancient and chic, dignified, colorful, vivid, simple, varied and changeful, graceful, profound, delicate, diverse, solid color and checkered pattern, beautiful and profound, strong and durable |
| Theme | Mysterious, realistic, lively, artistic, attractive, chic, auspicious, totem worship, inheriting ancient times |
| Pattern | Abstruse, unique, complicated, fine, kitsch, orderly, concise, chic, regular, neat, full, clear, compact |
| Color | Bright, Plain, Bright-colored, gorgeous, intensive, abundant, passionate, elegant, unified, exaggerated, bright and gorgeous |

### 4.3.2. Screening of Descriptive Vocabulary of Zhuang Brocade

According to Table 3 and expert opinions, 10 Zhuang brocade description vocabulary pairs with opposite meanings were obtained through group discussion and summarization, which include simple and complex, ancient and gorgeous, plain and colorful, orderly and disorderly, bright and dull, abundant and single, delicate and crude, beautiful and vulgar, classic and modern, ordinary and special. Then, the semantic differential method proposed by American psychologist Osgood was used in the questionnaire. Proposed by the American psychologist Osgood in 1957, the semantic differential method conducts perceptual experiments using the semantics of words. The semantic differential method can measure the emotional content of a word more objectively, which makes it one of the basic methods of emotional quantification in kansei engineering [23]. The basis of the semantic differential method is a series of adjectives and their antonyms. According to the semantic differential method, pairs of Zhuang brocade description adjectives with opposite meanings were placed at the left and right ends of the attitude scales in the questionnaire. The attitude scales were divided into 5 levels of feelings to measure the subjects' attitudes towards the description words of the 10 core Zhuang brocade samples. The purpose was to get more accurate data on consumers' cognition, attitudes, and preferences of the Zhuang brocade cultural traits. Table 4 shows the semantic differential test scale of sample Z1. The 10 groups of words correspond to 10 core samples and 10 semantic differential test scales.

**Table 4.** The semantic differential test scale of sample Z1.

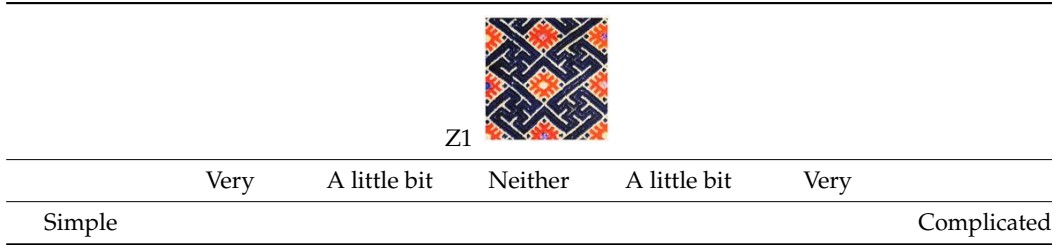

Z1

| | Very | A little bit | Neither | A little bit | Very | |
|---|---|---|---|---|---|---|
| Simple | | | | | | Complicated |

According to Tables 3–5, the Zhuang Brocade Cultural Traits Research Questionnaire was made. The purpose of the questionnaire survey was to extract the core vocabulary of Zhuang brocade cultural traits. Respondents to the questionnaire were mainly Chinese individuals aged 20 to 50, with knowledge of Zhuang brocade and an undergraduate education or above. Among them, 50% were from the Guangxi Zhuang Autonomous Region, and the other 50% were from tourists visiting Guangxi, people interested in ICH, and people interested in Zhuang brocade or cultural and creative products derived from ICH. Through field visits and online questionnaires, a total of 437 valid questionnaires were collected. The questionnaire has two parts, one of which is multiple-choice questions, as shown in Table 4. The other part is subjective questions, requiring the respondents to describe the cultural traits for each of the 10 core samples according to their subjective feelings and fill in the vocabulary. Through the questionnaire, consumers' awareness and

preferences of Zhuang brocade cultural traits were understood. Finally, based on the word frequency data, 10 core vocabularies most representative of the Zhuang brocade cultural traits were obtained, which corresponded to the 10 core samples, as shown in Table 6. The selection criterion was the most frequent and most appropriate description of the cultural traits of the sample.

**Table 5.** Descriptive pairs of Zhuang brocade.

| Sample | Descriptive pairs | Sample | Descriptive pairs |
|--------|-------------------|--------|-------------------|
| Z1 | Simple and complex | Z6 | Abundant and single |
| Z2 | Ancient and gorgeous | Z7 | Delicate and crude |
| Z3 | Plain and colorful | Z8 | Classic and modern |
| Z4 | Orderly and disorderly | Z9 | Beautiful and vulgar |
| Z5 | Bright and dull | Z10 | Special and ordinary |

**Table 6.** Shows the description vocabulary of 10 core samples and their corresponding cultural traits.

| Sample | | Cultural trait | Sample | | Cultural trait |
|--------|--|----------------|--------|--|----------------|
| Z1 |  | Concise and orderly | Z6 |  | Ancient and elegant |
| Z2 |  | Solid color and checkered pattern | Z7 |  | Plain and chic |
| Z3 |  | Varied and changeful | Z8 |  | Unique and delicate |
| Z4 |  | Bright and gorgeous | Z9 |  | Colorful and beautiful |
| Z5 |  | Classic and chic | Z10 |  | Elaborate and fine |

### 4.3.3. The Relationship between Descriptive Vocabularies of Zhuang Brocade and Core Samples

The 10 pairs of words correspond to 10 core samples. The third questionnaire survey was conducted with the 5-point Likert scale to obtain customers' cognition and psychological expectations of the cultural traits of Zhuang brocade more accurately and to quantify cultural emotions more scientifically. The consistency of the 10 core samples with the cultural trait descriptive vocabularies of the cultural traits of Zhuang brocade was scored. The five points are: very consistent (2), consistent (1), generally consistent (0), slightly inconsistent (−1), totally inconsistent (−2), as shown in Figure 3.

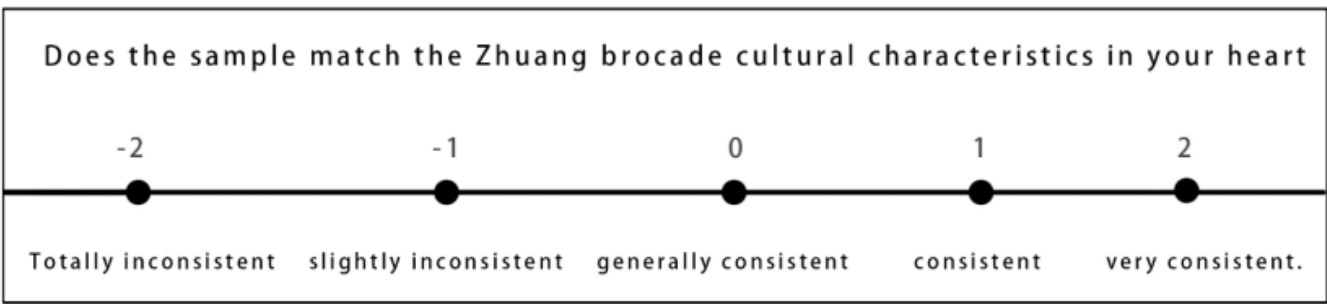

**Figure 3.** Does this sample conform to the cultural trait of Zhuang brocade in your mind?

Statistics: The questionnaire survey included 20 expert users (craft artists, designers, and ICH inheritors) and over 400 general users. A total of 376 valid questionnaires were obtained, and SPSS was used for reliability analysis. The Cronbach's alpha was 0.931, proving the good reliability of the questionnaire. Each core sample had 402 response scores (376 questionnaires × 10 pairs of descriptive vocabularies). The results were averaged as the score of the sample in terms of consistency with the descriptive vocabulary. For example, the score of Sample Z1 under "Simple and Complex" is 1.35, indicating that the visual perception of Sample Z1 is "Complex", and this perception is strong. Sample Z1 tends to have the cultural trait of "Complex". A negative number means that the trait is not prominent. The description vocabularies correspond to the three largest scores were adopted to represent the cultural traits of the sample. The scores of each sample in terms of consistency with the descriptive vocabularies are shown in Table 7.

**Table 7.** The scores of each sample in terms of the consistency with the descriptive vocabularies.

| The Number of the Sample | Descriptive Vocabularies of Cultural Traits | Score | The Number of the Sample | Descriptive Vocabularies of Cultural Traits | Score |
|---|---|---|---|---|---|
| $Z_1$ | Simple | 1.32 | $Z_6$ | Bright | 1.723 |
| | Orderly | 1.25 | | Delicate | 0.921 |
| | Neat | 0.97 | | Vivid | 0.905 |
| $Z_2$ | Beautiful | 1.88 | $Z_7$ | Elegant | 1.142 |
| | Colorful | 1.42 | | Plain | 1.231 |
| | Profound | −1.16 | | Fresh | 1.357 |
| $Z_3$ | Ancient | 0.981 | $Z_8$ | Orderly | 0.896 |
| | Classic | 1.432 | | Concise | −0.792 |
| | Gorgeous | −0.893 | | Bright | −0.817 |
| $Z_4$ | Plain | −0.917 | $Z_9$ | Delicate | 1.391 |
| | Bright-colored | 1.136 | | Bold | −1.37 |
| | Colorful | 1.425 | | Pretty | 0.514 |
| $Z_5$ | Mysterious | 0.804 | $Z_{10}$ | Orderly | 1.917 |
| | Bright-colored | 0.831 | | Chic | 0.302 |
| | Plain | 0.031 | | Beautiful | 1.659 |

Ten pairs of descriptive vocabularies reflect different visual perceptions of the samples. In terms of the correlation, if the absolute value of the correlation coefficient between the descriptive vocabularies is greater than 0.5, there is a significant correlation between the corresponding vocabularies. The Kaiser-Meyer-Olkin (KMO) test of descriptive vocabularies of Zhuang brocade was conducted. The KMO test statistic is an index mainly used in

factor analysis of multivariate statistics to compare the simple correlation coefficient and partial correlation coefficient between variables. The KMO statistic takes the value between 0 and 1. When the sum of squared simple correlation coefficients among all variables is larger than the sum of squared partial correlation coefficients, the KMO value is closer to 1, indicating that the correlation between variables is stronger and the original variables are more suitable for factor analysis. When the sum of squared simple correlation coefficients among all variables is closer to 0, the KMO value is closer to 0, indicating that the correlation between variables is weaker and the original variables are less suitable for factor analysis [24]. The KMO coefficient we obtained was 0.793, greater than 0.5. The Bartlett test chi-square value was 270.741. The degree of freedom was 81, and the *p*-value was 0.000. It is indicated that the survey results are suitable for factor analysis. The cumulative contribution of each factor is shown in Table 8.

**Table 8.** Cumulative factor contribution.

| Factor | Characteristic Value | Contribution Rate/% | Accumulative Contribution Rate/% |
|:---:|:---:|:---:|:---:|
| 1 | 4.901 | 51.231 | 50.0178 |
| 2 | 2.598 | 20.319 | 80.7214 |

It can be seen from Table 7 that there are two principal factors with a characteristic value greater than 1, and the cumulative contribution rate is 80.599%. The loss of factor interpretation is small. They can cover the most descriptive vocabularies of Zhuang brocade. In order to understand the composition of each principal factor, the maximum variance method was used to orthogonally rotate the factor loading matrix, and the factor loading matrix after rotation is shown in Table 9. The load of a factor reflects the relationship between the variable and the factor. A large absolute value of the load corresponds to a large absolute value of the relationship between the variable and the factor and a significant correlation between the variable and the factor.

**Table 9.** Factor loading matrix after rotation.

| Descriptive Vocabulary Pair of Zhuang Brocade | Element | |
|:---:|:---:|:---:|
| | Factor 1 | Factor 2 |
| Simple & Complex | 0.112 | 0.528 |
| Ancient & Gorgeous | −0.091 | 0.216 |
| Plain & Colorful | 0.103 | 0.391 |
| Orderly & Disorderly | 0.142 | −0.312 |
| Bright & Dull | 0.498 | 0.003 |
| Abundant & Single | 0.197 | −0.109 |
| Delicate & Crude | 0.185 | −0.312 |
| Beautiful & Vulgar | 0.176 | 0.175 |
| Classic & Modern | 0.181 | −0.214 |
| Ordinary & Special | −0.135 | 0.523 |

### 4.3.4. Quadrant Analysis of Descriptive Vocabularies of Zhuang Brocade

Quadrant analysis is a method to summarize the common causes of things with the same characteristics by attribution analysis [25]. According to the cultural characteristic descriptive phrases corresponding to the two components in Table 9, Factor 1 and Factor 2 can be summarized as: "Colorful" and "Plain". Then the quadrant analysis was applied. The score coefficients of each factor were obtained with regression analysis, and the linear relationship between cultural traits and descriptive vocabularies was obtained. The "plain

and ancient" was regarded as the *X*-axis, and "simple and colorful" was the *Y*-axis. Through factor analysis in SPSS, the quadrant distribution of the cultural traits of the samples was determined with regression analysis. The cultural traits of Zhuang Brocade were divided into four quadrants, including "Colorful", "Concise", "Ancient", and "Plain", as shown in Figure 4. Cultural traits presented by the Zhuang brocade can be divided into different quadrants. The quadrant name, sample attributes, and the scores of Zhuang brocade cultural trait descriptive vocabularies are shown in Table 10.

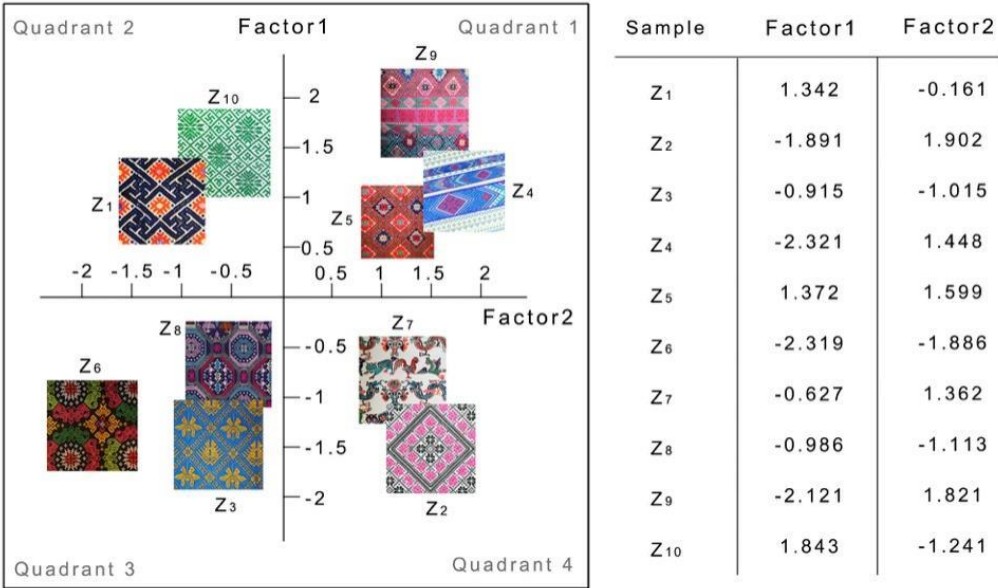

**Figure 4.** Quadrant distribution and the score of factors of cultural traits of Zhuang brocade.

**Table 10.** Scores of cultural trait vocabularies of Zhuang brocade samples.

| Quadrant | Quadrant Name | Sample | Descriptive Phrase | Score | Quadrant | Quadrant Name | Sample | Descriptive Phrase | Score |
|---|---|---|---|---|---|---|---|---|---|
| 1 | Colorful | $Z_4$ | Solid color and checkered pattern | 5.421 | 3 | Ancient | $Z_3$ | Varied and changeful | 5.901 |
| | | $Z_5$ | Classic and chic | 4.923 | | | $Z_6$ | Plain and chic | 4.812 |
| | | $Z_9$ | Colorful and beautiful | 5.901 | | | $Z_8$ | Unique and delicate | 4.039 |
| 2 | Concise | $Z_{10}$ | Elaborate and fine | 5.471 | 4 | Plain | $Z_7$ | Plain and chic | 5.723 |
| | | $Z_1$ | Concise and orderly | 4.172 | | | $Z_2$ | Solid color and checkered pattern | 4.027 |

## 5. Eye Tracker Experiment Based on the Cultural Traits of Zhuang Brocade

### 5.1. Experimental Design

#### 5.1.1. Experiment Purpose

The eye tracker experiment aims to verify the accuracy of the descriptive vocabularies of the cultural traits of Zhuang brocade obtained through the desktop research and questionnaire survey in the previous stage based on the four quadrants. Emotion, which is invisible and difficult to express, is very difficult to measure and describe the visual perception of Zhuang brocade carries its cultural traits [26]. Therefore, the eye movement experiment can present the subjects' rational choices when viewing Zhuang brocade more intuitively. The eye movement tracking experiment was carried out in a behavioral analysis laboratory. The eye-movement experiment ensures maximum rationality since it taps into the cultural traits based on the most direct reactions of the subjects by means of technology

and engineering. The experiment focuses on the area of interest (AOI) of the sample and the relationship between the registration time and the number of choices of samples [27]. The subjects of this experiment have almost no personal emotion for Zhuang brocade, and therefore cultural traits can be obtained in a more scientific and rational way.

### 5.1.2. Recruitment of Subjects

Since the conditions of subjects (age, education, occupation, income) could be variables affecting the experiment, 10 experts with research backgrounds related to Zhuang brocade were invited, including craft artists, designers, and ICH inheritors. The remaining 20 subjects were students of different professional backgrounds in colleges and universities. The subjects were composed of 15 females and 15 males. They were aged between 18 and 30 (ICH) information is attractive for young and middle-aged people), with naked or corrected visual acuity of 1.0 or higher and without color blindness. Regarding the electronic image reading behavior, all subjects have the experience of reading electronic images with mobile phones or tablet computers. No subjects saw the 10 core samples before the experiment [28].

### 5.1.3. The Process of the Experiment

The experiments were conducted in the behavioral analysis laboratory with Mobile Eye-XG Eye Tracking Glasses, a portable eye-tracking device developed by the Applied Science Laboratory (ASL) in the United States. In order to avoid the effects of some variables on the experimental data, various control parameters were set in the experiment, as shown in Table 11.

**Table 11.** The setting of variable factors in the eye tracker experiment.

| Variable | Influence Factor | Solution |
|---|---|---|
| | Viewing order | The application of Latin square design |
| | Visual persistence | The setting of the gray transition page |
| Experiment sample | Image restoration degree | The scanning of publications, real shots in museums, real shots of samples |
| | Size dimension | Pixels of $1800 \times 1800$ (px) or $1800 \times 2400$ (px), resolution of 300, RGB color mode |
| | Color distribution | The comparison between the brilliance group and color group |

In order to study the influence of color variables in Zhuang brocade on subjects' perception and judgment, a brilliance group was set corresponding to the color group. The brilliance group and the color group were shown to the subjects alternately, and the fixation time of each picture was 10 s.

Four tests of the four quadrants mentioned above were conducted. The samples for Test 1 are Z4, Z5, and Z9, all of which have the characteristic of "Colorful"; the samples of Test 2 include Z10, Z1, and Z4, and they all have the characteristic of "Concise"; the samples for Test 3 are Z3, Z6, and Z8, all of which have the characteristic of "Ancient"; the samples of Test 4 are Z7 and Z2, both of which have the characteristic of "Plain". In order to quantitatively evaluate the subjects' attention to the samples, each sample needs to be divided to get an independent AOI. After completing the test in one quadrant, the subjects rested for 10 min. The eye movement indicators include the recorded fixation time, fixation point, and the number of fixation. Fixation time indicates the sum of the subjects' overall interest in the sample, and it can reflect the subjects' attention to the sample. During the experiment, the project team verbally explained the experimental task to the subjects, briefly introduced the patterns, colors, pattern combinations, and theme of the Zhuang brocade. During the test, sample pictures were shown to the subjects who wore eye trackers by the screen. The receivers on the four corners of the screen were

used as reference coordinates. The eye trackers were aligned with the coordinates by the 9-point calibration.

### 5.2. Data Analysis of Eye Movement Experiment

The measurement indexes of the eye tracker in this experiment are the fixation time, the number of fixation points, and the lingering time. The fixation time is the average duration at all points, and it represents the degree of attraction of the concerned object. The longer the average fixation time, the higher the degree of concentration. In this experiment, based on the heat map and fixation time in AOI, the sample areas in different quadrants during the observation and selection of subjects were analyzed. Every subject had an AOI heat map in each quadrant. The heat maps of the 30 subjects in the same quadrant were stacked, and the heat map of this quadrant was obtained. It can show the visual attention of the subjects more intuitively. The heat map reflects the subject's fixation range and fixation time on the sample. The higher the color saturation, the higher the degree of attention. The larger the area, the wider the scope of attention.

Figure 5a shows the heat map and fixation time of Quadrant 1 "Colorful". It can be seen from the heat map that the subject's sight is relatively concentrated, and AOI is relatively large. As the pattern changes, the theme of the pattern becomes the visual focus. The PASS software was used for sample verification on the fixation time and number of choices. The results showed that significance = 0.0047, $p < 0.05$, and correlation = 0.992, indicating that fixation time has a positive relationship with the characteristic "Colorful", and Zhuang brocade has the cultural trait of "Colorful". Combining the first fixation time and fixation duration, we found that the three samples in Quadrant 1 have the longest first fixation time and overall fixation duration compared with samples in other quadrants. It is indicated that the cultural trait of "Colorful" is very prominent, and the samples in this quadrant are the most attractive to viewers.

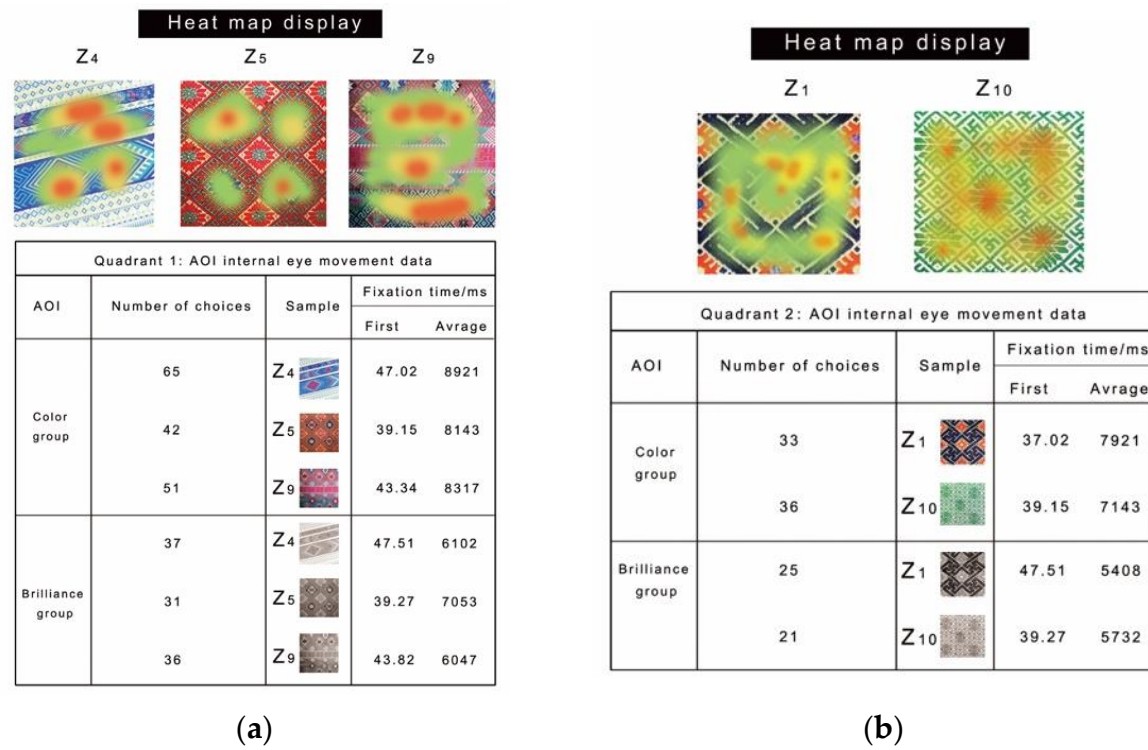

(**a**)　　　　　　　　　　　　　　　　　(**b**)

**Figure 5.** This is the heat map and fixation time data for quadrant 1 and 2. They should be listed as: (**a**) Heat zone map and gaze time data for quadrant 1 "Colorful". (**b**) Heat zone map and gaze time data for quadrant 2 "concise".

Figure 5b shows the heat map and fixation time of Quadrant 2 "Concise". It can be seen from the heat map that the subjects were more interested in the patterns of Sample Z2

and Sample Z5. The fixation time and the number of selections were verified with the PASS software. The results showed that the significance = 0.0037, $p < 0.05$, and correlation = 0.987, suggesting that the fixation time and the characteristic of "Concise" present a positive relationship, and Zhuang brocade have the cultural trait of "Concise". The feelings of subjects vary significantly. Combining the first fixation time and fixation duration, we found that the difference between the two samples in this quadrant was not significant, indicating that there was the cultural trait of "Concise". The overall attention degree of this quadrant was not high.

Figure 6a shows the heat map and fixation time of Quadrant 3 "Ancient". It can be seen that the sight of the subjects was relatively scattered, and the AOI was large but scattered. The fixation time and the number of choices were verified with the PASS software. The results showed that significance = 0.0039, $p < 0.05$, and correlation = 0.837, indicating that there is a positive relationship between fixation time and characteristic of "Ancient" and Zhuang brocade has the cultural trait of "Ancient". Combining the first fixation time and fixation duration, we found that the data of the three samples in this quadrant were quite different. Sample Z8 has the highest while Z3 has the lowest degree of attention. However, the overall degree of attention is good. It is indicated that there is a cultural trait of "Ancient", but the attraction of samples in Quadrant 3 to different viewers varies.

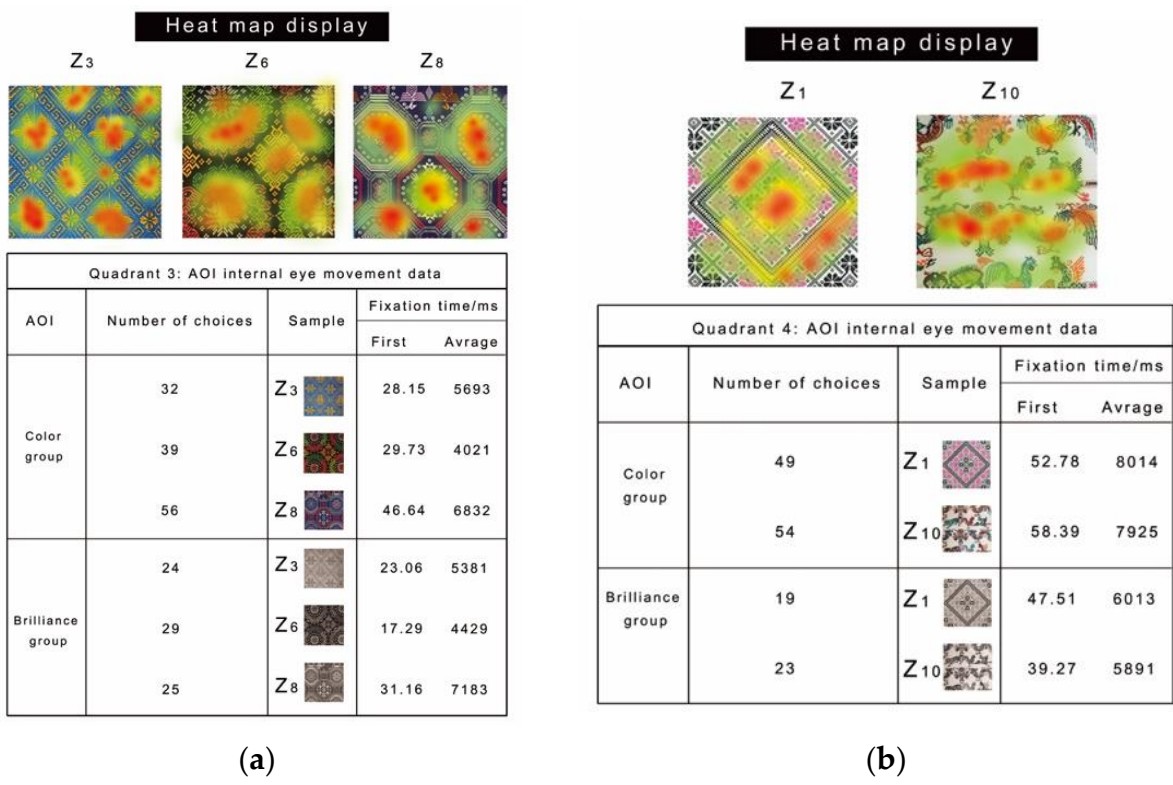

(**a**)                    (**b**)

**Figure 6.** This is the heat map and fixation time data for quadrant 3 and 4. They should be listed as: (**a**) Heat zone map and gaze time data for quadrant 3 "Ancient". (**b**) Heat zone map and gaze time data for quadrant 4 "Plain".

Figure 6b shows the heat map and fixation time of Quadrant 4 "Plain". It can be seen that the subjects' sight is relatively balanced, and they did not show interest in a certain area of a certain sample. The PASS software was used to perform sample verification on the fixation time and number of choices. The results showed that significance = 0.0041, $p < 0.05$, and correlation = 0.872, indicating that the fixation time has a positive relationship with the cultural trait of "Delicate", and Zhuang brocade has this cultural trait. Based on first fixation time and fixation duration, we found that the sample data are not significantly different. The feelings of different viewers do not vary significantly. The first fixation time

and fixation duration of samples in Quadrant 4 are higher than those in other quadrants, indicating that the viewers' attention degree is higher.

## 6. Analysis of Cultural Traits of Zhuang Brocade and Conclusion

By analyzing the heat map of the eye tracker experiment and the overall fixation data, the accuracy of the descriptive vocabularies of the cultural traits of Zhuang brocade can be verified according to the performance of the subjects when viewing samples in different quadrants. It was found that the patterns of Sample Z4 and Sample Z9 in Quadrant 1 attracted the most attention of subjects, indicating that they collectively reflect the cultural traits of Zhuang brocade to a certain extent.

Samples Z2, Z4, Z7, and Z10 are all white brocades but belong to different quadrants. The patterns of the Samples Z2, Z4, and Z7 are all checkered geometric figures. However, Z10 is a single-color plain brocade, so it more conforms to the cultural trait of "Concise". The patterns of Z2, Z4, and Z10 include moiré, octagonal shape, water ripple, 卐 and concentric circle. These patterns are combined with one another in these samples. The composition of geometric patterns is in single, two sides continual, and four sides continual forms. Since the geometric patterns are clear, bright, orderly, and neat, in the experiment, the difference between Z2, Z4, and Z7 lies in color. The patterns of Z10 are complex because they are animal motifs. But its overall color is plain, so it has a high degree of attention in Quadrant 4, where the characteristic of "Plain" is prominent.

The samples in Quadrant 3 are all "Gorgeous and Profound" brocades. They use bold colors, with red, yellow, blue, and green as the basic colors and other colors as complementary colors. Most of these samples use red and orange as the background and are embellished with contrasting colors such as blue and green. They also use blue and green as the background, embellished with red and orange. Their overall decorative effect is strong. Therefore, in Test 3, subjects focused on the patterns of samples, especially Sample Z8. Its heat map shows the largest response area and the most intense hot spots, indicating that Sample Z8 received the most attention, and it is more in line with the subjects' understanding of Zhuang brocade.

From the performance of Sample Z10 and Sample Z1 in the eye tracker experiment, it can be seen that in Quadrant 2, the focus is Zhuang brocade with geometric patterns as the main visual effect. The geometric pattern of this kind of Zhuang brocade is characterized by the staggering of large and small graphs, the interspersion of squares and circles, and the rigorous and regular layout. The colors of this type of brocade are mainly white embellished with blue, green, or black matched with yellow and orange. The uncomplicated color matching and the geometric shapes with a strong sense of order make the cultural trait of "Concise" particularly prominent in this series of Zhuang brocades. According to the experimental data, the cultural traits of "Plain" and "Concise" are presented in a different way, and this difference is related to the theme of Zhuang brocade. The cultural characteristic of "Plain" is mainly reflected in the samples of plant and animal themes. There are more curves in the patterns of Zhuang brocade with a plant theme. The curves in the patterns of Zhuang brocade with a flower theme make the cultural trait of "Plain" even more prominent. Most Zhuang brocades with the cultural characteristic of "Concise" have abstract geometry motifs or text motifs, and the most representative one is plain brocade.

From the performance of Sample Z4, Sample Z5, and Sample Z9 in the eye tracker experiment, it can be seen that the cultural characteristic of "Colorful" is prominent in Zhuang brocade with abundant color. This trait is mainly shown in colorful brocades. This type of Zhuang brocade is characterized by bold colors and complicated patterns. In complex patterns, there are seven to ten colors. Red, yellow, blue, and green are usually used as the basic colors, and other colors are complementary colors. The overall color contrast is strong, and the levels are rich.

From the data of Sample Z3, Sample Z6, and Sample Z8 in the eye tracker experiment, it can be seen that the Zhuang brocade cultural trait of "Ancient" is also prominent. This trait is represented in plain brocade. Plain brocade is often found in the headscarves

of Zhuang women. Zhuang women's headscarves are mainly woven with black and white cotton threads, with white cotton threads on the bottom and flowers woven with black cotton threads. The patterns are mainly flowers, with chrysanthemums being the most common. The cultural characteristic of "Ancient and Plain" is very prominent in plain brocade.

Through eye tracker experiments and comprehensive data, we can identify Z7, Z8, Z4, and Z10 as samples with the most typical Zhuang brocade cultural traits. The vocabularies, including "simple/unique/concise/beautiful/gorgeous/orderly" can accurately describe the cultural traits of Zhuang brocade. The results are shown in Figure 7.

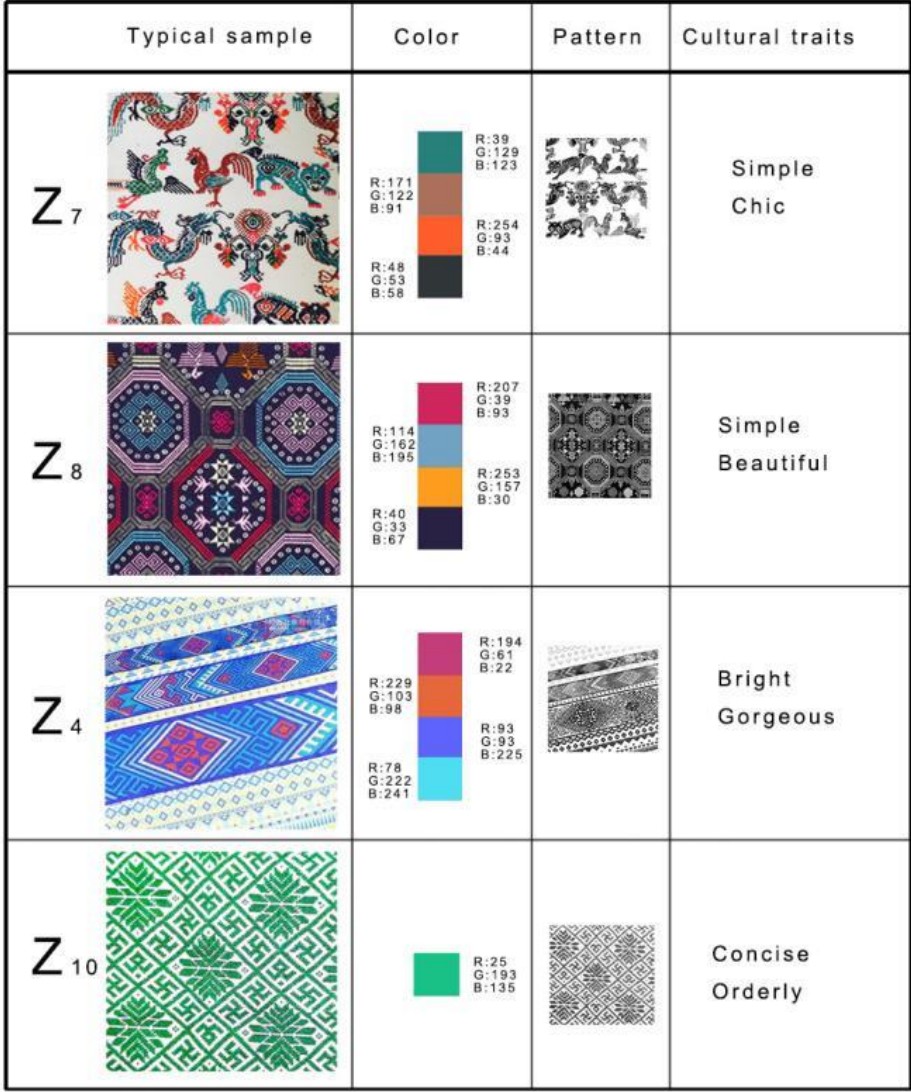

**Figure 7.** The most typical patterns and descriptive vocabularies extracted from the cultural traits of Zhuang brocade.

## 7. Conclusions and Prospects

This paper investigates the way to extract the cultural traits of ICH and explores the visual perception and emotional quantification of Zhuang brocade. We extracted and excavated the cultural traits of ICH by perceptual engineering, which is effective for emotional quantification [29]. Zhuang brocade is a highly visualized ICH. There is a strong connection between the viewer's visual perception and the cultural traits of the Zhuang brocade. In this study, based on a large number of sample images of Zhuang brocade, a picture sample database was established after screening these images according

to appearance variables. Desktop research, expert interview, and questionnaire survey were conducted to obtain data on the relationship between descriptive vocabularies of cultural traits of Zhuang brocade and the samples. The extraction of descriptive vocabularies helps to describe and determine the cultural traits of the Zhuang brocade. The cultural traits of Zhuang brocade were classified into four quadrants: "Colorful", "Concise", "Ancient", and "Plain". The eye movement experiment was combined to extract visual responses of cultural traits in the core samples. Through the analysis of the patterns of key areas, the representative pattern characteristics and color characteristics of the cultural traits of Zhuang brocade were obtained.

The theoretical contribution of this paper is as follows: First, the study of perceptual engineering is extended to the digital level of ICH. Second, a new research method is proposed for the emotional quantification of visual perception of Zhuang brocade. Third, a novel ICH data collection method is used. The visual response of the Zhuang brocade is obtained with the eye tracker. Fourth, the extraction of cultural traits of Zhuang brocade provides a theoretical basis for the modern application of Zhuang brocade and the development of derivative products of Zhuang brocade [30].

The practical value is, firstly, the methods applied in the above studies have a reference value for the extraction of the cultural traits of other ICH projects. The extracted cultural traits promote the inheritance of ICH and contribute to the understanding of ICH. Secondly, emotional quantification in perceptual engineering is applied to the emotional quantification of ICH with eye movement experiments. Eye-tracking technology has been widely used in many fields, such as the human visual system, psychology, cognitive language, and product semantics, to record users' eye movements during information interaction [31]. Through data analysis, this paper studies the law of visual perception of the subjects to understand the cultural traits of Zhuang brocade, which helps to understand Zhuang brocade more rationally. The subsequent research will apply the extracted cultural traits of Zhuang brocade to the design of ICH products. In the working environment of living inheritance of ICH, through market-oriented commercial mechanisms, ICH cultural products, which conform to contemporary taste and combine skills and cultural connotations, can be developed [32]. The economic and social benefits can, in turn, promote the expression of the cultural value of ICH in a new form on the basis of ICH preservation and inheritance and the protection and sustainable development of ICH. In the early stage of design, qualitative research and quantitative analysis of cultural traits of ICH will be conducted with perceptual engineering, which will help to correlate with the product qualities at a later stage of design [33]. In this way, modern cultural and creative products of ICH that meet consumer expectations and achieve high satisfaction can be designed. Additionally, the living inheritance of ICH and the benign combination of ICH with modern life can be achieved.

**Author Contributions:** Conceptualization, Y.H. and Y.P.; methodology, Y.H.; software, Y.H.; validation, Y.H.; formal analysis, Y.H.; investigation, Y.H.; data curation, Y.H.; writing—original draft preparation, Y.H.; writing—review and editing, Y.H.; visualization, Y.H.; supervision, Y.H. All authors have read and agreed to the published version of the manuscript.

**Funding:** This research received no external funding.

**Institutional Review Board Statement:** This study was approved by the Institutional Review Board (IRB) at the kookmin University of Technology. Informed Consent Statement: Informed consent was obtained from all participants involved in the study.

**Data Availability Statement:** All data generated or analyzed during this study are included in this article. The raw data are available from the corresponding author upon reasonable request.

**Acknowledgments:** The authors would like to thank all the participants in this study for their time and willingness to share their experiences and feelings. The authors would like to thank all the participants in this study for their time and willingness to share their experiences and feelings.

**Conflicts of Interest:** The authors declare no conflict of interest concerning the research, authorship, and publication of this article.

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
