# Peer review of "Discovery and Extraction of Cultural Traits in Intangible Cultural Heritages Based on Kansei Engineering: Taking Zhuang Brocade Weaving Techniques as an Example"

_applsci, doi:10.3390/app112311403_

Round 1

Reviewer 1 Report

The current study attempts to delve into the substance and appeal of Chinese intangible heritage drawing as an example Zhuang brocade weaving. I found the paper to be poorly written, and did not feel confident about the implementation of the experiments. The manuscript lacks structure, while the background and results of the study are not properly presented. I found the study to not have a clear and specific aim, and came away with too many questions to be able to recommend this paper for publication.
     I have several significant concerns about the presentation, research design, and general results that should be addressed prior to potential resubmitting for review.
     First, the title of the article may be misleading. I could not find any connection between the results and ‘Kansei engineering’ in the manuscript. Additionally, there is no ‘discovery’ of ‘cultural characteristics’, as these characteristics are, rather arbitrarily, assigned. 
     The overall presentation is poor. There are many hard to read sentences and grammar errors. At times, the text resembles oral speech or essay writing, and not a scientific manuscript. Furthermore, the scientific language used in not appropriate and should be revised. There are some errors with table and figure captions, and their contents are not adequately described in-text.
    A Background/Relevant Work section is completely missing. The authors should expand on the principles of scientific analysis methods used, and on the application for intangible heritage studies.
    The aims of the study are vague. The authors mention (lines 62–65) three very broad objectives, then (lines 76–81) two different objectives, and on subsection 4.1.1 the purposes of the second experimental part that do not necessarily align with the above. The scope of the paper should be stated clearly and solidly founded on the analysis of relevant work, in order to provide with specific research questions and prove the novelty of conducted research.
    The collection of the necessary dataset (section 3.1) is not adequately described, especially the selection of photographed objects. The authors also mention the collection of text, but that does not reflect into the rest of the manuscript. The ‘appearance’ variables and the criteria for their selection are not explained. The allocation of digits according to visual characteristics is difficult to understand. In my opinion the selection of participants for the experiments in 3.1, 3.2, 3.3, and 4.1 contains bias and is not adequately described. A description of the design of the questionnaires is missing. 
    A Discussion and Conclusions section is missing, where the authors could describe their interpretation of the surveys, focusing on the interrelation between visual perception, common patterns and colors on Zhuang brocades and, perhaps, intangible cultural heritage value.
    Most importantly, the declaration of research ethics is missing. Since the current study involves surveys, questionnaires, and images of human subjects the authors should check the research and publications ethics guide of the journal and provide a comprehensive report and documentation on the research ethics of their experiments.

Author Response

Dear reviewer, thank you for your patience to read and correct. None of your questions have been revised one by one, and there are no clear explanations. I have also explained them in the new manuscript. I'm sorry, but I only reply now. Because my mother died of cancer last month. I am her only child and I need time to deal with her affairs. So I am only correcting it now. Thank you for your understanding. Thank you for reviewing it again.

Reviewer 2 Report

This is a well-conceived research project. The goals and stages of the research are well-described from the gathering of samples (desk, literature, field research) through the questionnaires, eye-tracking, and analysis. A few things would be helpful. 

Please clarify the way in which they analysis of these formal brocade patterns fits within the framework of analyzing "intangible" cultural heritage. The brocades seem very tangible. If you are suggesting that the cultural values associated with them are the intangible cultural heritage, just state that clearly. 

Let us know whether the results were surprising in any way or whether they conformed to expectations?

It would be helpful to the reader to have a few terms defined clearly. On page 4, the variables should be paraphrased (e.g. what is "Vein" referring to as a variable?). 

A few technical terms could be made more clear. What is the EPrime software and what does it do? What is the KMO test coefficient?

Also, giving dates in Western time values for Han Dynasty and Ming Dynasty would help readers situate these brocades historically. 

The figure numbered Figure 5 on page 13 is perhaps actually Figure 8? 

{The English will need some attention. It is comprehensible, but has some grammatical errors.

The point about how the concept of "intangible cultural heritage" is being defined and used here probably needs attention and clarification. 

The question of whether or not the results were in any way surprising could be addressed to make clear whether all of this elaborate statistical, social science, and eye tracking actually contributed anything new to the understanding of these brocades. That is the one missing element in the paper. All of this elaborate work is well documented--but was it worth it? Do we need these techniques or do they merely affirm what was already known}

Author Response

(The authors gave the same response as above.)

Reviewer 3 Report

I find very interesting the article, which studies the historical-cultural characteristics and decorative modules of Zhuang brocade based on Kansei Engineering. It is a significant example of technology applied to the knowledge and preservation of the world cultural heritage.

Author Response

Thank you for your review. Thank you even more for your recognition. I will continue this project and do more in-depth research. Thank you for your support of my research. Thank you very much.

Round 2

Reviewer 1 Report

The authors have thoroughly addressed all of the concerns raised by the reviewer and corrected their manuscript appropriately. The manuscript has been restructured to adhere to the journal's instructions and the reviewer's suggestions. The introduction and background sections have been corrected, while all unclear points regarding the methodology have been addressed. Missing information about the conducted surveys have been added. The reviewer is confident that the manuscript will appeal to the journal's audience in its current form. The research is presented cohesively and comprehensively. The interpretation of the results and the derived conclusions are sufficiently backed up by the evidence provided. Therefore, the reviewer recommends that the manuscript gets accepted in the present form.

Author Response

Thank you for your recognition and support, thank you for your patience, and thank you for your efforts in this regard.